# Identification of Emerging Multidrug-Resistant *Neisseria gonorrhoeae* Isolates against Five Major Antimicrobial Agent Options

**DOI:** 10.3390/medsci11020028

**Published:** 2023-03-31

**Authors:** Sinethemba Hopewell Yakobi, Ofentse Jacob Pooe

**Affiliations:** School of Life Sciences, Biochemistry, University of KwaZulu-Natal, Durban 3629, South Africa

**Keywords:** antimicrobial resistance, antibiotics, *N. gonorrhoeae*, minimum inhibitory concentrations, multi-drug resistance, public health

## Abstract

Antimicrobial drug resistance in *Neisseria gonorrhoeae* has been documented all over the world. However, the situation in Sub-Saharan Africa has received little attention. It is critical to establish diagnostics and extend surveillance in order to prevent the emergence of illnesses that are resistant to several treatments. Monitoring antimicrobial susceptibility is critically required in order to gather data that may be utilised to produce treatment recommendations that will result in effective therapy, a decrease in *gonorrhoeae*-related difficulties and transmission, and effective therapy. Government authorities may set research and preventive objectives, as well as treatment recommendations, using data from the Gonococcal Antimicrobial Surveillance Program (GISP). Local and state health authorities may use GISP data to make choices about the allocation of STI prevention services and resources, to guide preventative planning, and to disseminate information about the most successful treatment practices. Using molecular and culture approaches, we investigated the occurrence of antibiotic resistance in isolates from KwaZulu Natal, South Africa. The great majority of gonococcal isolates (48% showed absolute resistance to ciprofloxacin), with penicillin and tetracycline resistance rates of 14% each. Only one of the gonococcal isolates tested positive for azithromycin resistance, with a minimum inhibitory concentration (MIC) of 1.5 µg/mL. Ceftriaxone was effective against all gonococcal isolates tested.

## 1. Introduction

The World Health Organization (WHO) predicted that 87 million new *Neisseria gonorrhoeae* (*N. gonorrhoeae*) infections occurred in people aged 15 to 49 in 2016 [1]. Sub-Saharan Africa was found to have the highest prevalence of *N. gonorrhoeae* [2]. Due to the development of resistance to every antimicrobial medication proposed for treatment since the introduction of sulphonamides in the 1930s, *N. gonorrhoeae* is a significant public health problem worldwide and is included on the WHO global priority list of antibiotic-resistant bacteria [3,4,5].

The recent discovery of gonococcal strains resistant to ceftriaxone and azithromycin in Australia and the United Kingdom has raised concerns about the possibility of untreatable *N. gonorrhoeae* [6,7,8]. Furthermore, a lack of knowledge regarding the antibiotic resistance spectrum of *N. gonorrhoeae* strains circulating in Sub-Saharan Africa, the region with the highest infection frequency, raises concerns about the spread of this incurable *N. gonorrhoeae* [9]. The lack of access to laboratory diagnostic facilities and the use of syndromic care in the treatment of sexually transmitted infections (STIs) are the two key reasons for antibiotic resistance data scarcity in Sub-Saharan Africa [10,11]. The use of syndromic management has a number of drawbacks, the most significant of which include a lack of susceptibility testing, an inability to recognise silent infections, limited options for extensive monitoring, and a lack of data on patients who have failed therapy [9,12]. In South Africa, 1 g of azithromycin and 250 mg of ceftriaxone are often used to treat syndrome-related diseases [13].

Men who have sex with other men (MSM) and those who have recurring episodes of *N. gonorrhoeae* have been identified as key transmission groups in the development of treatment resistance [9]. Several studies demonstrate that MSM in South Africa and other African nations have a high burden of *N. gonorrhoeae* infections; the first two instances of cefixime-resistant *N. gonorrhoeae* infections in Africa were documented in the MSM community [9,14]. Understanding the antibiotic resistance profile of gonococcal populations in core transmission groups and those under sentinel monitoring is therefore critical for influencing clinical treatment recommendations and policy planning [2,15]. Surveillance is critically crucial for detecting and tracking gonococcal resistance. In most clinical settings, *N. gonorrhoeae* culture is not frequently accessible [16]. Nonculture-based diagnostic procedures, such as nucleic acid amplification tests (NAATs), are increasingly being used in more nations, and antibiotic susceptibility testing is not regularly conducted in most laboratories [17,18]. In these contexts, *N. gonorrhoeae* infections are not reported together with drug susceptibility, and other monitoring approaches are used. As a consequence, the Centers for Disease Control and Prevention (CDC) undertook yearly large-scale investigations to evaluate the antibiotic susceptibility of *N. gonorrhoeae*, such as the National *Gonorrhoeae* Therapy Monitoring Study [13,19]. The purpose of this research is to report on the resistance patterns and trends of *N. gonorrhoeae* to specific antimicrobials in isolates obtained from the University of KwaZulu Natal.

## 2. Methods

Sixty-four *N. gonorrhoeae* isolates were donated by the University of KwaZulu Natal, Medical Microbiology. The isolates were grown from urethral swabs and cultured on Chocolate Agar and subsequently kept at −80 °C. These isolates were inoculated onto New York City (NYC) agar and were incubated for 24 h at 37 °C in 5% CO_2_. After the incubation time, we performed the Gram stain, oxidase test, catalase test, and the NAAT on the isolates to confirm *N. gonorrhoeae* identification. The pure culture isolates were suspended in trypticase soy broth containing 20% (*v*/*v*) glycerol and immediately frozen at −80 °C.

### 2.1. Antibiotic Resistance

Antibiotic resistance toward ciprofloxacin, azithromycin, ceftriaxone, penicillin G, and tetracycline was identified using the disc diffusion technique and Etests (bioMérieux, Midrand, South Africa). The minimum inhibitory concentrations (MICs) were interpreted in accordance with the breakpoints established by the European Committee for Antimicrobial Susceptibility Testing (EUCAST), with the exception of azithromycin, for which the epidemiological cut-off (ECOFF) value was used due to the lack of a resistance breakpoint. Azithromycin is always combined with another effective drug. The ECOFF for testing purposes to identify acquired resistance mechanisms is 1 µg/mL [20]. The ATCC 49926 *N. gonorrhoeae* strains are used for quality control.

### 2.2. Data Analysis

Antimicrobial susceptibility trends were assessed based on the Clinical and Laboratory Standards Institute (CLSI). For the ceftriaxone and azithromycin antibiotics, for which the CLSI has not established resistance criteria, the surveillance definition of reduced susceptibility was applied to isolates with MICs that are above the wild-type distribution (ceftriaxone MIC 0.125 µg/mL and azithromycin MIC 2.0 µg/mL). The significance criterion for bivariate analysis of epidemiological features was established at less than 0.05.

## 3. Results

### Antimicrobial Susceptibility Testing

All gonococcal cultures were positively identified as *N. gonorrhoeae* via the molecular technique, NAAT. All the target isolates demonstrated some level of phenotypic resistance toward at least two target antibiotics and thus were classified as multidrug-resistant (MDR) *N. gonorrhoeae*. Antimicrobial susceptibility data associated with *N. gonorrhoeae* resistance are presented in Table 1.

Most of the isolates were phenotypically resistant to ciprofloxacin, *n*  =  60 (94%), followed by penicillin *n*  =  29 (45%) and tetracycline *n*  =  28 (44%), respectively. Azithromycin resistance, interpreted using ECOFF values, was identified in 1/64 (2%) gonococcal isolates (MIC 1.5 µg/mL). There was no resistance or decreased susceptibility towards ceftriaxone among our isolates. Only three of the isolates (4.7% of the total) were susceptible to all of the target antibiotics, while 13 of the isolates (20.3%) were resistant to at least one of the target antibiotics, the majority of which (92%) were resistant to ciprofloxacin and 8% accounted for penicillin resistance. The majority of the isolated *N. gonorrhoeae*, 26 isolates (41%), were found to be resistant against two of the target antibiotics. All 26 of these isolates were primarily resistant to ciprofloxacin (100%) and of these, 12 isolates (46.2%) presented with a co-resistance to penicillin and 14 isolates (53.8%) presented with a co-resistance to tetracycline. Resistance against tetracycline and ciprofloxacin constituted the majority of co-resistance in these clinical isolates. Resistance against 3 of the target antibiotics was observed in 21 isolates; all these isolates conferred resistance against ciprofloxacin, tetracycline, and penicillin. Only one isolate, ISID 26, managed to confer resistance against 4 of the target antibiotics, namely ciprofloxacin, tetracycline, penicillin, and azithromycin.

Absolute resistance to the target antibiotics used in this study was found in 62 instances in these clinical isolates, with 9 of 29 counts of penicillin resistance (31%), 11 of 28 counts of tetracycline resistance (39%), and 32 of 60 counts of ciprofloxacin resistance (53%). Only five MDR isolates exhibited absolute resistance against these three antibiotic drugs (ISID 5, 7, 21, 45, 59), while seven isolates expressed absolute tetracycline and ciprofloxacin resistance, and 11% of the isolates exhibited absolute penicillin and ciprofloxacin resistance. On the other hand, only four isolates (6%) exhibited absolute tetracycline and penicillin resistance. The isolate that was shown to confer resistance to azithromycin exhibited with absolute resistance to ciprofloxacin and tetracycline; however, penicillin resistance was reported at an MIC of 8 µg/mL.

Only the relationship between penicillin-resistance and tetracycline-resistance development in these clinical isolates exhibited a covariance value of 60.6 and a correlation value of 0.43, with a lower 95% confidence interval (CI) of 0.2 and an upper 95% CI of 0.6, and a *p*-value of 0.0004, indicating statistical significance. The mean of penicillin resistance was 7.7 g/mL with a standard deviation (SD) of 11.8; similarly, the mean of tetracycline resistance was 8.1 g/mL with an SD of 11.9. It was determined that the fit mean was 7.7 with a standard error (std error) of 1.4. As a summary of the fit, the RSquared value (the coefficient of determination) was seen to be 0.19, but the lack of fit had an F-Ratio of 0.74 and the probability > F = 0.72. The parameter estimates for the intercept were observed at 4.2, with a standard error of 4.2. We also applied the t-value to measure the difference between the antibiotic-resistance means and found the t-ratio to be 2.6 with a probability greater than t equal to 0.012 (see Figure 1). Bivariate fit revealed that there was greater correlation that reflected the link between the two variables and indicated a potential link between the development of penicillin and tetracycline resistance (with the strongest linear relationship between the two variables).

Other relationship associations failed to show any statistical significance, according to the *p*-values, in the connection between the emergences of resistance in clinical isolates of *N. gonorrhoeae*. With a lower 95% confidence interval of −0.05, an upper 95% CI of 0.42, and a *p* value of 0.12, the covariance between penicillin and ciprofloxacin resistance was 32.8 and the correlation was 0.2. The mean level of ciprofloxacin resistance was 19.6 µg/mL, with a standard deviation of 14.1. The R-Squared value for the fit was 0.03, while the F-Ratio and prob > F values for the lack of fit were 0.3 and 0.12, respectively. The intercept’s parameter estimations were 4.5 with a 2.5 standard error. To measure the difference between the antibiotic-resistance means, we utilised the t-value, and we saw that the t-ratio was 1.8 with a prob > t = 0.082 (see Figure 2). Tetracycline and ciprofloxacin resistance were linked with a covariance value of 21.8 and a correlation value of 0.13. The lower 95% confidence interval was 0.11 and the upper 95% was 0.36. The R-Squared value for the fit was 0.02, whereas the lack of fit presented an F-Ratio of 1.8 and prob > F = 0.07. With a 2.1 standard error, the parameter estimates for the intercept were 18.4. To measure the variation in antibiotic-resistance means, we performed the t-value, and we found that the t-ratio was 8.6 with a statistically significant prob > t = 0.001.

When comparing two of the three antibiotic resistances (penicillin versus tetracycline versus ciprofloxacin), the two variables that are compared are positively related, and they move in the same direction. The data that we have presented show that the covariance values have increased, which demonstrates to a significant degree that this is the case. The correlation coefficients were used as a tool for determining the degree to which two variables had a linear connection at a certain juncture in time. We were able to detect the direction and degree of antibiotic-resistance connection with the assistance of a correlation coefficient interval chart. First, a very weak correlation of 0.13 was observed between tetracycline resistance and ciprofloxacin resistance; second, a weak positive correlation was observed between penicillin resistance and ciprofloxacin resistance; and third, a moderately positive correlation was observed between penicillin resistance and tetracycline resistance among the isolates used in this study.

## 4. Discussion

Antimicrobial resistance in *N. gonorrhoeae* is a global health issue because it increases the likelihood of untreatable *N. gonorrhoea* [21]. Based on our results, it is plainly clear that greater attention must be devoted to *N. gonorrhoeae* infections, particularly in low-resource settings such as the South African public health care system. All of the *N. gonorrhoeae* strains investigated were classed as MDR bacteria. The majority of the isolates had high rates of ciprofloxacin resistance, followed by tetracycline, penicillin, and azithromycin resistance, respectively. In South Africa, syndromic treatment for STIs has enabled these drugs to be used for longer periods of time [13,22]. Similar considerable rates of antimicrobial resistance for these target drugs have been documented, consistent with data from national surveillance as well as the results of more recent studies done in the South African province of Gauteng [23], which our research (94% resistance rate) mirrored. Ciprofloxacin was recently removed off the market owing to disturbingly high rates of resistance [24]. It is conceivable that this is attributable to the use of ciprofloxacin in the treatment of male dysuria [25], as well as the continued use of ciprofloxacin by certain medical centres for the treatment of male urethral discharge [9]. At the current moment, it is not viable to repurpose this medicine for use in the treatment of syndromic diseases due to the persistently high resistance rate [26,27]. It was discovered that ciprofloxacin resistance mutations occurred over time in the portions of the gyrA and parC genes that govern quinolone resistance [28]. The high incidence of tetracycline resistance is thought to be caused by the use of doxycycline as part of the syndromic therapy for non-gonococcal urethritis [29]. The tetM plasmid and RpsJ V57M polymorphisms are found in the majority of tetracycline-resistant isolates [29]. Furthermore, several isolates have been shown to have both the GGI, a type 4 secretion system linked with the propagation of AMR to numerous antimicrobials among gonococcal species, and plasmid-mediated AMR [30]. A significant prevalence of GGI was discovered in a Kenyan surveillance survey [29]. The researchers discovered that 94% of gonococcal strains have both GGI and plasmid-mediated AMR in their genomes [29]. In South Africa, urethral discharge was not treated with azithromycin until 2015; before that year, the antibiotic was not widely available in the public health system [31]. Despite the fact that azithromycin has only been used to treat STIs for a short period, studies done in South Africa discovered that 15% of MSM isolates were azithromycin resistant [9]. Only one of the 64 isolates tested positive for resistance to this antibiotic, with a MIC of 1.5 g/mL, and was classified as “resistant” using the EUCAST breakpoint. Because azithromycin is often used in conjunction with other medications for treatment, EUCAST no longer serves as an azithromycin susceptibility breakpoint; instead, ECOFFs are utilised to define azithromycin resistance. This highlights the need to include male groups, notably MSM, in the process of routine drug screening. At this moment, it is unclear if azithromycin resistance is limited to core group populations or has extended to the broader population in South Africa. National surveillance among symptomatic persons at sentinel sites showed a modest incidence (3%) of *N. gonorrhoeae* azithromycin resistance [13]. This monitoring, however, does not explicitly cover key transmission groups. Current data from two clinics in the KwaZulu Natal area, on the other hand, suggest the opposite; they discovered azithromycin resistance in 68% of isolates [31]. The MIC was measured in this investigation utilising agar dilution methods rather than the Etest. Nonetheless, all these studies together provide strong evidence that increased surveillance of azithromycin resistance is critical. This is because newly established resistance may impair the effectiveness of syndromic treatment. In a lot of countries, ceftriaxone is now recommended as the first-line therapy for *N. gonorrhoeae* [19]. Even though the azithromycin and ceftriaxone combined therapy was recommended by WHO, a reevaluation of this therapy may be required for treating *N. gonorrhoeae* [8,32]. In situations of syndromic treatment, dual therapy with azithromycin and ceftriaxone is recommended and has been shown to result in levels of macrolide consumption that can exceed and overcome resistance thresholds [33]. In this regard, the fact that no isolates were found to be resistant to ceftriaxone is promising.

Although azithromycin and ceftriaxone resistance was identified to be insignificant in this study, it is worth noting that antimicrobial resistance of *N. gonorrhoeae* to the presently available first-line monotherapy, ceftriaxone, has become an impending danger to efficient gonorrhoea control in China. From 2013 to 2016, surveillance data gathered by the China Gonococcal Antimicrobial Resistance Surveillance Program (China-GRSP) revealed a high prevalence (varying between 9.7% and 12.2%) of 3827 isolates with reduced susceptibility to ceftriaxone (MIC 0.125 mg/L) [34]. Since then, it has been recorded in Japan [35], Australia [36], Canada [37], Denmark [38], Ireland [39], the United Kingdom [40], and Singapore [41]. Shigemura and his colleagues likewise reported that a deletion mutation in the mtrR promoter region is a plausible mechanism of azithromycin resistance and may be a major predictor of increased MICs (0.5 g/mL or more) in *N. gonorrhoeae* infection [42]. A recent study found that gonococci had a lower antimicrobial sensitivity to azithromycin, which may reduce the drug’s efficacy [43]. As previously observed, the lower azithromycin susceptibility resulted from a variety of processes. Antibiotic resistance has developed in tandem with their widespread usage in medicine and animal agriculture.

In view of the expected epidemic of MDR *N. gonorrhoeae*, it is critical that sexual health care services be improved throughout the country. Intensified clinical governance and antimicrobial stewardship, the introduction of molecular diagnostics, the careful selection of empirical treatment regimens, the evaluation of new potential drugs like zoliflodacin [44], and investment in an enhanced antimicrobial surveillance structure that includes core transmission groups are all required to avert such a development and a Sub-Saharan Africa epidemic of untreatable *N. gonorrhoeae*.

This study has several limitations, the most prominent of which being the small number of isolates assessed. Secondly, there is an inability to comprehend high-risk groups and core transmission groups due to a lack of patient information. Despite the small number of isolates studied, our findings provide an overview of the range of antibiotic resistance of *N. gonorrhoeae* strains circulating in KwaZulu Natal, which is consistent with other published reports. To further understand the mechanism of resistance posed by the 64 isolates investigated, whole genome sequencing will be undertaken.

## 5. Conclusions

Monitoring *N. gonorrhoeae* antibiotic susceptibility has the ability to improve knowledge regarding gonococcal resistance, influence policy and preventive actions, and provide data on which to base national treatment recommendations on a regular basis. We were able to discover azithromycin resistance in our investigation, which might be the start of a trend that would considerably complicate the treatment of *N. gonorrhoeae*. Local and national health agencies may use this information to determine how to allocate services and resources for the prevention of STIs, as well as to influence the planning of preventative and control initiatives. Continued monitoring, appropriate treatment, the development of novel medications, and transmission prevention remain the most effective strategies for lowering the incidence and morbidity of *N. gonorrhoeae*.

## Figures and Tables

**Figure 1 medsci-11-00028-f001:**
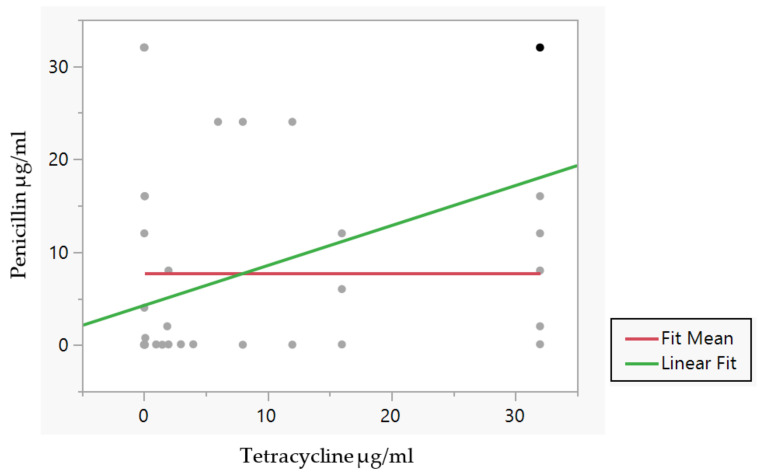
Bivariate fit of penicillin resistance by tetracycline resistance in 64 clinical isolates of *N. gonorrhoeae*, with each point representing the antibiotic concentration needed to for efficient inhibition of isolates conferring resistance when correlating penicillin resistance and tetracycline resistance, created using JMP Pro 16.

**Figure 2 medsci-11-00028-f002:**
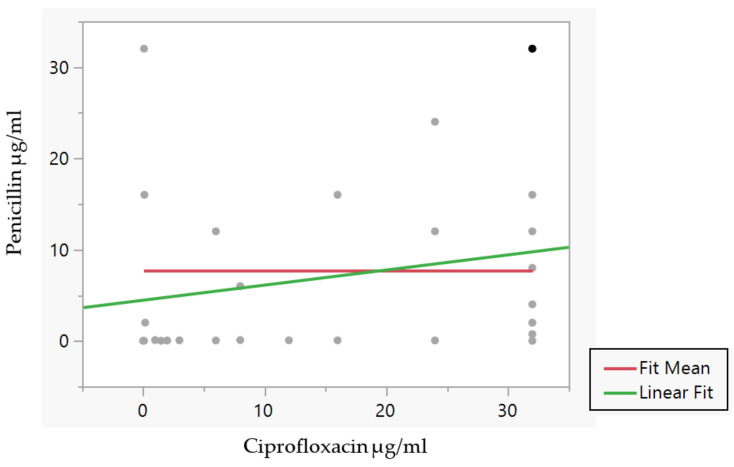
Bivariate fit of penicillin resistance by Ciprofloxacin resistance in 64 clinical isolates of *N. gonorrhoeae*, with each point representing the antibiotic concentration needed to for efficient inhibition of isolates conferring resistance when correlating penicillin resistance and ciprofloxacin resistance, created using JMP Pro 16.

**Table 1 medsci-11-00028-t001:** Antimicrobial susceptibility profiles and MICs for the *Neisseria gonorrhoeae* isolates (*n*  =  64).

Drug	Number of Isolates:	MIC (µg/mL)			
	Susceptible	Resistant	Median	Mean	Range
^a^ Ciprofloxacin	4	60	2	8	0.016–32
^b^ Azithromycin	63	1	0.094	0.2	0.016–1.5
^a^ Penicillin	35	29	0.094	3.7	0.016–32
^a^ Tetracycline	36	28	0.094	3.1	0.016–32
^a^ Ceftriaxone	64	0	0.006	0.012	0.002–0.12

^a^ EUCAST breakpoints were used to classify strains as susceptible or resistant. ^b^ ECOFF value used classify strains as susceptible or resistant.

## Data Availability

All data generated or analysed during this study are included in this published article.

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
