# Peer review of "Identification of Emerging Multidrug-Resistant Neisseria gonorrhoeae Isolates against Five Major Antimicrobial Agent Options"

_medsci, 2023, doi:10.3390/medsci11020028_

Round 1

Reviewer 1 Report

This is a nice report about multi-drug resistance in N. gonorrhoeae collected in South Africa. This is a small study of 64 isolates. It was unclear what this collection presented- there was no detail about where the strains came from in terms of region, gender, timeframe. Adding these details would have added  a lot of value to this descriptive study. It was disappointing that the strains were not sequenced or RT-PCR tests performed to back up the antibiogram data. From lines 130 to 175- I was unclear why this analysis was performed at all. Was this to try and judge weather co-administration of antibiotics was leading to resistance? Although the analysis was performed, there was no discussion about it, so it was unclear if it was in fact necessary for the paper. Typically antibiotic resistance markers have been long associated with certain genetic lineages, and so the prevalence of the resistance is related to importation of these lineages via travel rather than local treatment practices. A sentence or two on the bivariate fit data would be good in the discussion.

Author Response

This is a nice report about multi-drug resistance in N. gonorrhoeae collected in South Africa.

This is a small study of 64 isolates. This is due to the scarcity of samples and a very limited time that was allocated for collection.

It was unclear what this collection presented- there was no detail about where the strains came from in terms of region, gender, timeframe. Adding these details would have added  a lot of value to this descriptive study.  Some of these suggested details have been added, however due to ethical and confidential reason, not all details can be released as yet.

It was disappointing that the strains were not sequenced or RT-PCR tests performed to back up the antibiogram data. This will be explored as the study progress.

From lines 130 to 175- I was unclear why this analysis was performed at all. Was this to try and judge weather co-administration of antibiotics was leading to resistance? Although the analysis was performed, there was no discussion about it, so it was unclear if it was in fact necessary for the paper. No co-administration was performed, the gramma concerning this was addressed.

Typically antibiotic resistance markers have been long associated with certain genetic lineages, and so the prevalence of the resistance is related to importation of these lineages via travel rather than local treatment practices. A sentence or two on the bivariate fit data would be good in the discussion. Bivariate fit data is discussed.

Thank you very much for these suggestion that seek to better present the study.

Reviewer 2 Report

This Article discusses about the emerging multidrug resistant Neisseria at South Africa that would increase the awareness for the clinicians around the world.

It is written in a very detailed manner with data analysis and graphic representation. 

It would be appealing for readers if article also compares azithromycin susceptibility in different parts of the world where high resistance of azithromycin has been reported like in Hawaii, Greece, Cyprus. 

Author Response

It is written in a very detailed manner with data analysis and graphic representation. 

It would be appealing for readers if article also compares azithromycin susceptibility in different parts of the world where high resistance of azithromycin has been reported like in Hawaii, Greece, Cyprus. 

Data of azithromycin susceptibility and resistance around the world has been discussed, together with ceftriaxone susceptibility.

Thank you for the contributions towards the quality of the paper.

Reviewer 3 Report

This is an important piece of research work that will inform on the extent of AMR in gonococcal isolates from a region of the world with high disease transmission. Could the authors please consider making revisions to the manuscript using the below points as guide:

Line 11 - 12: Check grammar: successful therapy / effective therapy mentioned twice as benefit of gonococcal AMR monitoring

Line 67: What tme period were these isolates collected. A table with this data for each isolate (year, region of South Africa collected from, etc) would be good to see.

Line 68: "These isolates were, inoculated plates of ... " - please check grammar.

Line 68: What is NYC agar? Full description necessary as this is first mention in manuscript.

Line 81-82: Check grammar (plural issue)

Line 97: Table 1 not Tables 1

Line 99: Table 1 legend. Italicise Neisseria gonorrhoeae

Line 107: By target antibiotics, do the authors mean all other antibiotics except Azithromiycin and Ceftriaxone? Otherwise the sentence is incorrect as only 1 strain was reported resistant to Azithromycin and none to Ceftriaxone. Hence, impossible for 3 strains to be resistant to all antibiotics.

Discussion: Do the authors know the mechanism of resistance in any of the 64 isolates tested e.g. via sequencing of segments of the gyrA and parC genes mentioned in Line 194?

Author Response

Line 11 - 12: Check grammar: successful therapy / effective therapy mentioned twice as benefit of gonococcal AMR monitoring. Completed.

Line 67: What tme period were these isolates collected. A table with this data for each isolate (year, region of South Africa collected from, etc) would be good to see. Details of these have been added to the paper, however not all details could be entered as yet due to ethical and confidential matters, that will be addressed soon.

Line 68: "These isolates were, inoculated plates of ... " - please check grammar. Completed.

Line 68: What is NYC agar? Full description necessary as this is first mention in manuscript. Completed.

Line 81-82: Check grammar (plural issue). Completed.

Line 97: Table 1 not Tables 1. Completed.

Line 99: Table 1 legend. Italicise Neisseria gonorrhoeae. Completed.

Line 107: By target antibiotics, do the authors mean all other antibiotics except Azithromiycin and Ceftriaxone? Otherwise the sentence is incorrect as only 1 strain was reported resistant to Azithromycin and none to Ceftriaxone. Hence, impossible for 3 strains to be resistant to all antibiotics.  The sentence says that, out of the 64 isolates, 3 isolates had no antimicrobial resistance to any of the target drugs.

Discussion: Do the authors know the mechanism of resistance in any of the 64 isolates tested e.g. via sequencing of segments of the gyrA and parC genes mentioned in Line 194? This will be revealed as the study progresses, as WGS of the isolates will be underway.

Thank you so much for the contributions made to better the quality of the paper.